# Protocol: Using Single-Case Experimental Design to Evaluate Whole-Body Dynamic Seating on Activity, Participation, and Quality of Life in Dystonic Cerebral Palsy

**DOI:** 10.3390/healthcare8010011

**Published:** 2019-12-31

**Authors:** Hortensia Gimeno, Tim Adlam

**Affiliations:** 1Complex Motor Disorders Service, Paediatric Neurosciences, Evelina Children’s Hospital, Guy’s & St Thomas’ NHS Foundation Trust, London SE1 7EH, UK; 2Department of Psychology, Institute of Psychiatry, Psychology and Neurosciences, King’s College London, London SE5 8AF, UK; 3UCL Global Disability Innovation Hub, UCLIC, Department Computer Science, UCL at Here East, 8-9 East Bay Lane, Queen Elizabeth Olympic Park, London E15 2GW, UK; t.adlam@ucl.ac.uk; 4Designability Charity Ltd., Royal United Hospital, Bath BA1 3NG, UK

**Keywords:** dystonic cerebral palsy, single-case experimental design, N-of-1 trial, dynamic seating

## Abstract

*Introduction:* People with hyperkinetic movement disorders, including dystonia, experience often painful, involuntary movements affecting functioning. Seating comfort is a key unmet need identified by families. This paper reports a protocol to assess the feasibility and preliminary evidence for the efficacy of dynamic seating to improve functional outcomes for young children with dystonic cerebral palsy (DCP). *Design:* A series of single-case experimental design N-of-1 trials, with replications across participants, with a random baseline interval, and one treatment period (*n* = 6). *Methods*: Inclusion criteria: DCP; 21.5 cm < popliteal fossa to posterior sacrum < 35 cm; Gross Motor Function Classification System level IV–V; mini-Manual Ability Classification System level IV–V; difficulties with seating. *Intervention:* Trial of the seat (8 weeks), with multiple baseline before, during and after intervention and 2 month follow up. The baseline duration will be randomised per child (2–7 weeks). *Primary outcomes:* Performance Quality Rating Scale; Canadian Occupational Performance Measure; seating tolerance. The statistician will create the randomization, with allocation concealment by registration of participants prior to sending the allocation arm to the principal investigator. Primary outcomes will be assessed from video by an assessor blind to allocation. *Analysis:* Participant outcome data will be plotted over time, with parametric and non-parametric analysis including estimated size effect for N-of-1 trials.

## 1. Introduction

Dystonia is a neurological syndrome that differs from spasticity and it is characterised by involuntary, patterned, sustained, or repetitive contractions of opposing muscles, resulting in abnormal twisting body movements and abnormal postures [1].

Children with dystonia experience involuntary movements throughout their waking day, and this affects all areas of functioning defined in the International Classification of Function (ICF) [2]. Dystonic movements are often triggered by attempts to move [3], limiting the ability of the child to participate at the very moment they wish to do so. It is often painful and causes discomfort. This may lead to frustration and distress, which further compounds the severity of the involuntary movements, causing a toxic feedback loop. Gimeno et al. [4] reported and ranked functional concerns identified by families and young people with dystonic movement disorders: first was pain affecting function, and access to assistive technology; second, dressing, hygiene and handling tools (including feeding); and third, participation in outdoor activities. Access to assistive technology and pain were highlighted as the highest priority concern(s) for more than half the group. In their systematic review of interventions for children with cerebral palsy, Novak et al. [5] identified a pronounced lack of high-quality research in assistive technology to support activity and participation. Thus, the support of activity and participation is a high priority for work in this under-researched area.

Dynamic seating is a type of seating that enables the occupant to move while seated. Dynamic seats are used in a wide variety of contexts, including vehicles and offices. However, in this context, dynamic seating is designed to facilitate the movements of disabled people, while providing postural support. There are multiple dynamic seats on the market that are designed to be used by children with cerebral palsy, however they differ from the seat that is the subject of this trial. Some seats (the R82 x:Panda for example) simply have a sprung backrest, hinged about an axis close to the child’s hips, and are able to move backwards if leaned upon. Others, such as the Netti or Interco Aktivline have more complex mechanisms that enable the user to extend their hips and knees. There is little published research evaluating dynamic seating for people with dystonia [6,7].

The present seat provides postural support for children with dystonic cerebral palsy and whole-body involuntary extensor movements. It enables the occupant to independently extend their hips, knees and spine while remaining seated. The seat’s supports yield to powerful involuntary movements and support voluntary movements through pivots that are aligned with the occupant’s joints and are gently sprung. The principal weight-bearing support is a saddle on which the child sits. The seat consists of a chassis to which is attached two independently hinged leg supports and a hinged backrest. The child’s thighs are supported independently by moveable supports that are sprung and hinged on an axis aligned with the child’s hip joints. The child’s feet are supported by a pair of foot supports that are attached to the leg supports through hinged joints aligned with the child’s knees. Each foot support is sprung, able to move independently of the other foot, and pivots about an axis aligned to the ankle joint. The back rest provides support to the child’s back, the sides of the thorax, and the head. A 4-point pelvic belt, thigh straps, ankle harnesses and a chest harness prevent the occupant from falling out of the seat.

### 1.1. Dynamic Seating Research

Initial screening of papers for a current systematic review of dynamic seating [8] has yielded only two peer-reviewed scientific studies evaluating existing commercial dynamic seats [6,7]. These seats do not accommodate asymmetric movements, yet most involuntary dystonic movements are asymmetric, and cause unwanted postural displacement in seating that enforces symmetry.

In a consultation on the project, parents were strongly supportive of the trial of an already designed new seat. They liked the minimal visual impact of the seat and emphasised the need to design the seat for easy adjustment, cleaning, transfers and transport. In regards to methodology, parents suggested that a seat should be provided for each child, and that the quantity of written feedback from parents should be minimised.

### 1.2. Why a Trial Is Needed

In July 2012, the National Institute for Care and health Excellence (NICE) spasticity guidelines [9] (which includes other co-existing motor disorders such as childhood dystonia) reviewed the effectiveness of occupational therapy and/or physiotherapy interventions. Only 14 publications were included and out of those, only one study included children with dystonic/hyperkinetic phenotype [10], with the majority of the study including children with spastic CP.

Further, the guideline states that children with CP should have timely access to equipment necessary for their management programme; for example, postural management equipment, including sitting and standing systems. It states that intervention should be focused on enhancing skill development, function and ability to participate in everyday activities, as well as preventing adverse consequences such as pain or contractures. There is no previous research that evaluates the feasibility and acceptability of the dynamic chair and that this is imperative because present guidelines highlight enhancing skill development, function and ability.

In conclusion, children with dystonic movement disorders including dyskinetic CP, and their families report sitting comfortably as one of their major concerns and challenges [4]. Difficulties with sitting amplify their difficulties participating in all areas of life. A dynamic seat that offers flexible and comfortable support may facilitate inclusion and participation and promote a more positive motor-sensory feedback loop, potentially leading to improved function and quality of life.

The proposed study will be a proof of principle using single-case experimental design (SCED) to evaluate the feasibility and potential efficacy of using a dynamic seat in children with dystonic cerebral palsy to inform future clinical trials.

## 2. Aims and Objectives

### 2.1. Purpose of the Research

The overall aim of this work is to test the feasibility of using the proposed protocol for testing the potential efficacy of a specific dynamic seat for children with dystonic cerebral palsy, for the purpose of guiding the design of a larger efficacy trial.

This study will investigate the use of the dynamic seating system on a range of activity and participation outcomes for young children with severe dystonic CP, with particular interest in achieving parent-defined functional goals, activity, participation and quality of life.

### 2.2. Overarching Study Aim

To examine the feasibility and acceptability of a clinical trial investigating the efficacy of a dynamic seating system for young children with severe dystonic cerebral palsy. The evidence will help inform a future full-scale randomised clinical trial.

### 2.3. Specific Primary Aims per Series of N-of-1 Trials

Assess variability of outcome using N-of-1 type design (between- and within-period variances) and estimate effect size (for N-of-1 trials) of treatment outcomes to inform the design of future clinical trials.Evaluate the potential efficacy of the dynamic seating system intervention at varying times (across a number of weeks) using N-of-1 type design.Assess whether dynamic seating is associated with improved individual functional performance on (parent) identified goals as measured by the primary outcome measures.Assess whether changes can be observed in secondary outcomes, including health-related quality of life.

### 2.4. Hypothesis

The use of whole-body dynamic seating can improve comfort, activity, participation, and quality of life in young children with dystonic cerebral palsy.

## 3. Method

### 3.1. Design

The proposed study design is a series of N-of-1 trials, with replications across subjects (N-of-1 plus 5 replications). For each child, there will be at least 5 baseline assessments over at least 2 weeks in their usual seating (each trial provides one data point or baseline assessment, for example, reaching forward to hit a switch is one data point/one baseline). Therefore, there will be multiple baselines within each visit session. The baseline period will be followed by an intervention assessment period of 8 weeks when they are using the dynamic seat, and then a 2 month follow-up period. The duration of the baseline will be randomised for each child and will be between 2 and 7 weeks and therefore participants with longer baselines will have a larger number of baseline data-points. The baseline period will represent treatment as usual (TAU). In this way, comparison of TAU and dynamic seating provision is possible, with each participant acting as their own control.

### 3.2. Recruitment

Participants will be recruited through Evelina London Children’s Hospital, Guy’s and St Thomas’ National Health System Trust. Additional recruitment centres providing care for children with dystonia will be enrolled only if needed to reach the target numbers.

### 3.3. Eligibility

Inclusion criteria for the study are: (a) diagnosis of dystonic cerebral palsy; (b) 21.5 cm < popliteal fossa to posterior sacrum < 35 cm; (c) Gross Motor Function Classification System (GMFCS) levels IV–V [11,12]; (d) Manual Ability Classification System (MACS)/mini-MACS levels IV–V [13]; and (e) difficulties reported with current seating.

Exclusion criteria are: (a) pure spasticity or mixed phenotype when spasticity is the dominant feature; (b) dystonia due to a neurodegenerative condition; (c) scheduled for other surgical treatments (e.g., hip surgery) in the study period; (d) scheduled to receive botulinum toxin injections during the study period; and (e) epilepsy.

### 3.4. Withdrawal of Subjects

Premature withdrawal of an enrolled participant will be considered if the child/parent expresses reluctance or appears reluctant to engage in the study, even if consent was provided at the point of recruitment. Participants will be free to withdraw at any time without giving a reason for their decision and with no consequence to their clinical care. The right to withdraw will be explained verbally and in the information sheets.

### 3.5. Sample Size

In SCED or N-of-1 design with replications across participants, sample (series) size is not based on the power to test group effects using inferential statistics, as each separate trial examines change over time within an individual. This will allow us to determine whether treatment has an effect for an individual. In N-of-1 studies, the number of replications (participants recruited) chosen after the first case is often based on pragmatic grounds related to the known heterogeneity of the clinical sample, and the number of measurements within each time period is based on the likely variability in outcome. It is recommended that N-of-1 plus 3 replications are necessary as a minimum to explore the efficacy of an intervention, but a total of 5 replications is better. Similarly, it is recommended that at least 3 assessment points are measured at baseline. Multiple baseline assessments will be made per child per week during the randomised baseline period that varies between 2 and 7 weeks.

A sample size of 6 will be sufficient to provide information on whether the intervention is effective for individuals, to examine the direction of ‘average treatment effect’ across individuals, and to provide estimates of the within- and between-subject variability of outcomes in this population. It will also allow us to obtain information on ease of recruitment and adherence to protocol.

### 3.6. Randomisation and Allocation Concealment

Treatment allocation will take place by randomisation with participants of assigned baseline length. Allocation concealment will be achieved by registration of consented participants on to a randomisation database prior to the allocation arm being sent to the principle investigator via email, and from there to the centre delivering treatment.

The trial statistician who has no direct contact with the clinical aspects of the trial or the study team will perform randomisation. Block randomisation will be performed, and a list will be generated using Stata (https://www.stata.com/). The length of each participant’s baseline period will be randomised to a non-repeating sequence of random integer durations that are between 2 and 7 weeks, where 2 weeks ≤ baseline period ≤ 7 weeks.

### 3.7. Blinding and Blind Rating

Due to the nature of the intervention (a dynamic seat), children, parents and the visiting occupational therapist will not be blinded to the randomised baseline period allocation. The investigators will assess all children before randomisation takes place. Primary outcome measures will be administered and videoed by a member of the research team. Raters of these video-recorded assessments will be blind to the time of assessment.

Research therapists will assess all children at each assessment time point, using video, with subsequent ratings performed by trained blind assessors. Primary and secondary outcome measures will be videoed for subsequent rating by trained independent raters, blind to the time of assessment in relation to treatment. The order of video segments will be randomised before being collated for rating. Metadata, including date, will be removed from the videos to maintain blindness.

### 3.8. Intervention to Be Studied

The KiTE 2 whole-body dynamic seat is designed for children with dystonic cerebral palsy [14]. It combines postural support and maintains lateral spinal alignment in the child, while also allowing substantial extension of the hips, knees and ankles should the child experience an involuntary dystonic movement or choose to move voluntarily. The seat is able to accommodate asymmetric movements as each leg is supported and able to move independently of the other and of the backrest. There is no mechanism linking the movements of the backrest and leg supports. The joints of the seat are pivoted anatomically, with each pivot point in the seat being aligned with the pivot points of the child’s own joints.

### 3.9. Assessments

Baseline assessments and outcome measures will use multi-dimensional assessments across the ICF [2]. Measures of impairment, activity and participation will be used for sample description, outcome evaluation before and after intervention and at follow up two months after the conclusion of the intervention period. Assessments at intake will not be repeated post intervention. Validation of outcome measures in dystonia is limited with most assessments used in trials having been validated for other populations [15]. As this is a feasibility study, some of the feasibility parameters as per the National Institute for Health Research (NIHR) are: (i) to calculate the standard deviation of the primary outcome measure needed to estimate sample size on a future large scale trial and (ii) to explore characteristics of the proposed outcome measures [16]. Whilst some studies including children and young people with dystonia have explored the use of outcome measures such as as the Canadian Occupational Performance Measure (COPM) [17], and the Performance Quality Rating Scale [18], also proposed in this study [19,20] they evaluated neurosurgical and rehabilitation interventions. The possibility of applying those outcome measures to evaluate functional improvement in younger children with dystonia following provision of specialist equipment is unknown and this feasibility study will explore their application and whether changes to implementation are required. Reliability and Validity of these measures when applied to dystonic movement disorders can be therefore explored at posteriori.

### 3.10. Primary Outcomes

There will be three primary outcome measures. Two complementary co-primary objective outcome measures will be employed: the first one will be a subjective rating scale to explore parents/carers perceived performance and satisfaction on a number of self-selected goals using the Canadian Occupational Performance Measure (COPM) [17]. Secondly, the Performance Quality Rating Scale (PQRS) [18] will be used to measure the individual goals selected for the intervention and will be blind-rated. Further, seating tolerance will be objectively measured using sensors attached to the chair that will capture the duration and frequency of its use throughout the baseline and intervention and follow-up periods.

### 3.11. Secondary Outcomes

There will be four secondary outcomes. The first one, The Responsive Augmentative and Alternative Communication Style Scale (RAACS), will be used to measure interaction between a parent or carer, and the child at pre-intervention, post-intervention, and 2 month follow up. The second one, Paediatric Quality of Life Inventory (PedsQL), will rate the levels of difficulty in a number of areas. The third one will measure the upper limb reach. The fourth one will measure the sleep quality.

### 3.12. Baseline Assessments for Treatment Planning

One assessment will be used to establish goals for intervention: COPM [17]. This assessment will also be used as outcome measure

### 3.13. Other Demographic and Clinical Measures

Demographic measures (age, gender, etc.) will be used for descriptive purposes. The Burke–Fahn–Marsden Dystonia Rating Scale (BFMDRS) [21] will be used to evaluate dystonia severity following the published protocol.

Functional motor abilities will be classified using the Gross Motor Classification System (GMFCS) [12] and the MACS/mini-MACS [13,22]. These ordinal scales have been validated for use with children with CP and the levels provide information about limitations in daily life with higher scores implying more functional difficulties. The use of these scales for children with other conditions other than CP has been described as GMFCS and MACS equivalents. This does not require involvement of the children or families and it is a classification system that can be done after observing the children move and handle objects and through the information gathered on other assessments explained below.

### 3.14. Description of Standard Care

Participants will be randomised to length of baseline and the same measurements will be acquired during this period when the child will be using their usual static seating system (chair). As the provision of static seating tends to differ across the country, we will capture this data as part of the feasibility parameters captured. We will record other type of standard care that participants receive including occupational therapy, physiotherapy and any other early intervention.

### 3.15. The Seat Intervention

The specifically designed dynamic seating system will be set-up by a qualified occupational therapist/physiotherapist and an engineer from the study team. The intervention (dynamic chair provision) will consist of a trial of 8 weeks. Figure 1 shows the seat model that will be trialled. A researcher will visit the child and family once per week to measure goals set up by the family. This, in itself, could be considered an intervention as the family might not have set clear goals if they were not involved in the trial. These visits will be video recorded for subsequent scoring and review.

### 3.16. Inter- and Intrarater Reliability

Reliability training will be required for the independent assessors. For initial training, the raters use videos from the CMDS database and live assessments of a number of consecutive children. An interclass correlation coefficient ≥0.8 between raters and within raters over repeat tests will be required before a rater is able to continue. The ICC will be calculated for the primary outcome measure, the Performance Quality Rating Scale (PQRS) [18]. At the end of the trial, raters will be re-assessed on the training videos to ensure there has been no rating drift.

### 3.17. Assessment Time Points (Data Collection)

Table 1 shows the study timeline and participant journey for assessment time points and data collection. Figure 2a,b are flow charts describing the processes for the baseline and intervention periods in the study and should be read in conjunction with Table 1.

#### Outcome Evaluation

Primary and secondary outcome measures across activity and participation domains will be completed at baseline (b1–b7, with baseline length randomly allocated), during intervention (for repeated measures such as PQRS, sensor based data and accelerometers) (t1–t8), post intervention (p1, p2) and follow up (f1, f2) and include the COPM, PQRS, seat occupancy, RAACS, Peds-QL, time to target, path complexity, reaching performance and sleep quality.

Response will be tracked using multiple video recordings during sessions of goals identified for intervention as well as with sensor data and data from accelerometer. The Performance Quality Rating Scale (PQRS) will be used for goals, and sensor data for seat occupancy, and frequency and intensity of movement in bed as an indicator of sleep quality. Data points will be collected at baseline (b1–b7), during intervention (t1–t8); post-intervention (p1) and 2 month follow up (f2). There will be a minimum of 3 video-recorded baseline assessments of each goal before intervention and minimum of 3 trials post intervention and at 2 months review. Further, outcome evaluation measures outlined above will be completed at t1 and t2. The PQRS will be rated by an independent observer using video recordings. The videos from baseline, post-test and follow up will be presented in a randomized non-chronological order. A third party will complete assessments at baseline and post treatment. PQRS will be scored blinded by another occupational therapist using video recordings randomly presented.

### 3.18. Sensor-Based Measurements

**Seat occupancy profile:** This will be measured continuously at baseline, during and after the intervention periods, using a simple pressure switch and data logger attached to the participant’s seat. The data logger will record when and for how long the seat is occupied. The system requires no intervention from the participant or their family between setup and removal.

**Sleep quality:** This will be measured using a simple lightweight inertial measurement unit (IMU) attached to the child using a wrist or ankle band. This measurement has been trialed with children during the previous and preparatory KiTE 1 project [23]. Parents will be asked to ensure that their children wear the sensor band for at least 2 nights per week during the baseline and intervention periods. The sensors require no intervention by the parent other than placing the sensor on the child’s wrist or ankle and switching it on when they go to bed.

**Daytime activity intensity and frequency:** To be measured using the IMUs for 2 days per week, preferably two days either side of a measured night. The sensors will be used to measure how much the child moves, and the intensity and duration of their movement.

Sensor data collection and analysis is shown on Table 2. It will be collected by a study researcher during weekly visits to the participant and transferred to secure encrypted storage using the project laptop. Data will be coded and pseudonymised on collection.

### 3.19. Data Analyses Plan

A full statistical analysis plan for the study will be available. Descriptive statistics will be used to summarise demographic characteristics of the participants, including age and diagnosis. Functional ability will be classified using GFMCS and MACS/Mini-MACS.

Primary and secondary outcome measures across the domains of the ICF will be completed at baseline, post-intervention and follow up. There will be two types of data depending on whether the within- or between-subjects data are examined. Within subject data will evaluate the SCED data with direct replications. Between-subjects evaluation for outcome measures pre-intervention, post-intervention and follow up will be used for all measures calculating means and 95% confidence intervals for overall change. There is no planned interim analysis.

#### 3.19.1. Data from SCED

To describe the impact of the intervention in the single-case experimental studies, pre-intervention and post-intervention raw scores will be examined in all outcome measures. Analysis of PQRS changes will be evaluated using an appropriate analysis for SCED. The analysis will be guided by interpretation of the visual representation of the results; whether there is auto-correlation on baseline data; and whether there is baseline trend present in the data set. We will follow the recommendations from the Single-Case Reporting Guideline in Behavioural Interventions (SCRIBE) [24]. Visual analysis will be undertaken to evaluate changes in the domains of means, levels, trends, variability, latency and consistency. This will be followed by statistical analysis of the graph which will vary depending on serial dependency. Estimated effect size will be calculated using a suitable regression approach for SCED. We will initially fit a naïve linear regression model using ordinary least squares for reference only. We will use the Tau-*U* approach as it takes into account baseline trend and is considered one of the most robust effect size methods in SCED [25].

SCED uses repeated measures and therefore provides multiple data points. The extended baseline and intervention phases will result in many data points. PQRS can be applied multiple times within the same practiced task, given it is scoring the quality of performance rather than one-off task performance [20]. Therefore, each task step within a task is assigned a data point.

#### 3.19.2. Sensor Data Analysis

See Table 2 for a summary description of the data to be collected and the proposed derived parameters for further analysis.

### 3.20. Feasibility

Feasibility parameters in preparation for a main trial will be assessed and practicalities about the study delivery evaluated. These include completion and withdrawal rates, completion of questionnaires at baseline, post-intervention and follow up but also other outcomes such as management resources (i.e., monitor time for managing the trial and data, resources required for assessment and delivery of the intervention and time for data entry and data analysis. Scientific outcomes will also be captured as indicators of acceptability of the measures used.

## 4. Trial Status and Ethics Approval

Awaiting ethics approval. Not yet recruiting.

## 5. Patents

No patents applied for.

## Figures and Tables

**Figure 1 healthcare-08-00011-f001:**
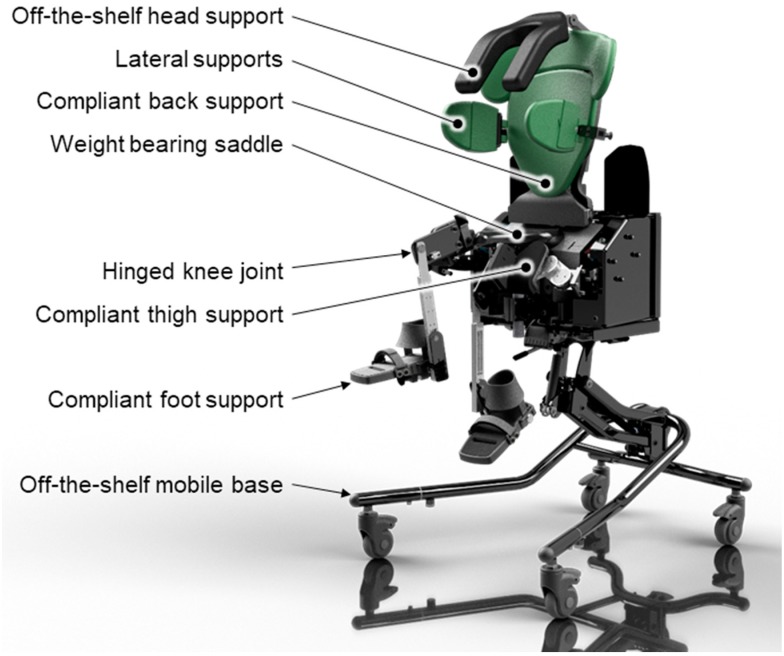
A computer model of the KiTE seat to be used for the study, showing the dynamic foot, thigh and back supports. The seat is mounted on a standard mobile base as used in classrooms and domestic environments.

**Figure 2 healthcare-08-00011-f002:**
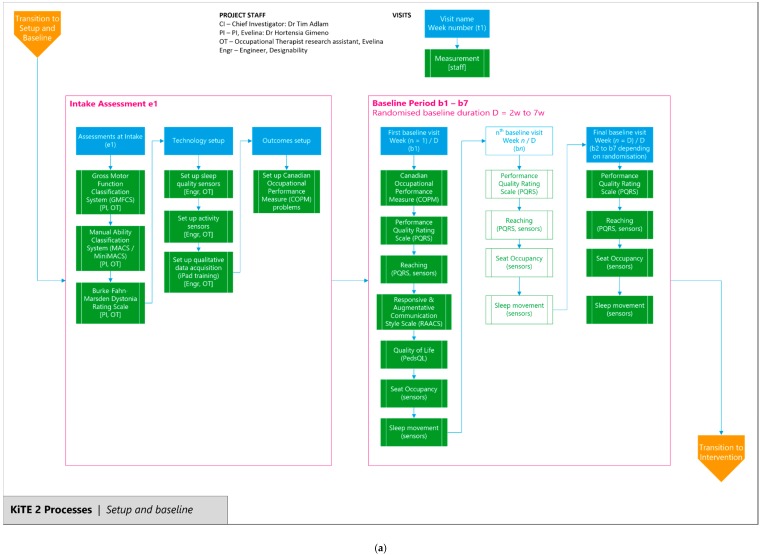
(**a**) Flow chart of baseline measurements and assessments. This chart should be read in conjunction with Table 1. (**b**) Flow chart of intervention period measurements and assessments. This chart follows Figure 2a and should be read in conjunction with Table 1.

**Table 1 healthcare-08-00011-t001:** Study assessments and participant journey timeline. The baseline and intervention processes are described with flow charts in Figure 2a,b.

	Enrolment	Assessments							Transition Visits (Configure Seat)		Post-Allocation								Review				Close Out
		Baseline									Intervention (8 Weeks)								Post		Follow Up		
		(Duration Randomised between 2 and 7 Weeks)																					
TIMEPOINT	e1	b1	b2	b3	b4	b5	b6	b7	c1	c2	t1	t2	t3	t4	t5	t6	t7	t8	p1	p2	f1	f2	co1
**ENROLMENT**																							
Participant Identification	X																						
Eligibility Screen	X																						
Participant Recruitment (Including Consent)	X																						
Randomisation (From b1–b2 to b1–b7)	X																						
**INTERVENTION**																							
Dynamic seat provision																							
**ASSESSMENTS**																							
GMFCS	X																						
mini-MACS/MACS	X																						
Demographics	X																						
BFMDRS (Dystonia)	X																		X				
**TIMEPOINT**	e1	b1	b2	b3	b4	b5	b6	b7	c1	c2	t1	t2	t3	t4	t5	t6	t7	t8	p1	p2	f1	f2	cc1
PQRS (Task Performance)		xxx	xxx	xxx	xxx	xxx	xxx	xxx			xxx	xxx	xxx	xxx	xxx	xxx	xxx	xxx	X				
COPM (Goal Attainment)		X																		X		X	
Seat Occupancy (Sensor)		X	X	X	X	X	X	X			X	X	X	X	X	X	X	X		X		X	
RAACS (Communication)		X																		X		X	
Sleep Quality (Sensor)		X	X	X	X	X	X	X			X	X	X	X	X	X	X	X					
QoL (PedsQL)		X																		X		X	
**ANALYSIS AND REPORTING**																							
Study Forms																							X
Study Files																							X
Study Findings																							X
Summary Letter																							X

Abbreviations: e: Enrolment, b: baseline, c: configuration of seat, t: treatment, p: post-intervention, f: follow up, co: close out, GMFCS: Gross Motor Function Classification System, MACS: Manual Ability Classification System, BFMDRS: Burke–Fahn–Marsden Dystonia Rating Scale, PQRS: Performance Quality Rating Scale, COPM: Canadian Occupational Performance Measure, RAACS: The Responsive Augmentative and Alternative Communication Style Scale, QoL: Quality of Life, and PedsQL: Paediatric Quality of Life. Note: X: indicates when assessment will take place. The intervention (dynamic seating provision) is represented with blocked colour across t1–t8.

**Table 2 healthcare-08-00011-t002:** A table of sensor data to be collected and the initial analyses to be applied.

Sensor	Measurement, Example Data Points	Analysis	Output Parameters
Seat occupancy	Time of seat entry Time of seat exit Date Time Data 2017-08-21; 12:33:45;1 2017-08-21; 12:33:45;0	Continuous seat occupancy profile: Identification of initiation time, duration and exit time of each seat occupancy event.	Weekly seat occupancy ratio (occupied time / unoccupied time)—for what proportion of the day is the seat used on average? Mean seat occupied session duration per week—how long is each session? Mean daily seat occupancy profile per week—when is the seat used? Seat entry frequency per week—How often is the seat used?
3 axis integrated accelerometer	Night Accelerations of the wrist and ankle ($a_{W}$ and $a_{A}$) Date Time x y z 2017-08-21; 12:33:45.67;123;456;243	Night Acceleration vector magnitude: $\left|a\right|=\sqrt{a_{x}^{2}+a_{y}^{2}+a_{z}^{2}}$ Times when acceleration vector magnitude is above and below sleep/awake threshold.	Night Sleep quality index—proportion of time in motion above threshold intensity—How well does the child sleep? Sleep time—total time per night not in motion above threshold intensity—For how much time is the child asleep each night?
	Day Accelerations of spine (vertebra T1) and left and right thighs during seat use: ($a_{S},\;a_{TL}\;and\;a_{TR}$)	Day Intensity of motion: time averaged amplitude of xyz acceleration vector time averaged amplitude of jerk (rate of change of acceleration)	Day Motion profile—How active is the child throughout the day? Maximum and minimum motion intensities—What are the maximum and minimum activity levels? When do they occur?
9 degree of freedom motion and orientation sensor	Movement of the child’s wrist during a reaching task. Measurement of the acceleration, rotation and geomagnetic field orientation of the child’s wrist. 3 axis acceleration (accelerometers) 3 axis rotation (gyroscopes) 3 axis geomagnetic field orientation (magnetometers)	Motion path of the child’s wrist derived from acceleration and orientation data	Output parameters are to be determined from analysis of the data in the context of the video and PQRS assessment. Parameters sought are to be representative of the ‘smoothness’ and efficiency of the child’s hand motion.

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
