# Peer review of "Protocol: Using Single-Case Experimental Design to Evaluate Whole-Body Dynamic Seating on Activity, Participation, and Quality of Life in Dystonic Cerebral Palsy"

_healthcare, 2019, doi:10.3390/healthcare8010011_

Round 1

Reviewer 1 Report

Well-designed and innovative study! 

Author Response

Thank you very much for your advice.

Reviewer 2 Report

Overall, my comments were address. I suggest making the limitations clearer (e.g. parent reported outcomes).

Author Response

Thank you very much for your advice. We have revised the manuscript accordingly.

Reviewer 3 Report

Thank you for your responses and incorporation of my previous suggestions into the revised manuscript. I had also suggested adding a flow-chart to the manuscript to increase clarity on timing and individual involved in completing the various assessments. I also suggested adding a brief statement summarizing reliability/validity and justification for your selected measures.

In the response, it states that a flow chart has been created for the manuscript and information on each measure added to this. I do not see a flow chart in the version of the manuscript that I have downloaded. I am unsure if this is an oversight, or if I have been given access to the wrong version of the manuscript!

Round 2

Reviewer 3 Report

Thank you for adding the flow diagram- It is excellent and increases the clarity for the reader. I believe this manuscript is ready for publication